# Pathogenic *FANCC* Variants Are Associated with Accessory Breasts in a Sub-Saharan African Multiplex Family

**DOI:** 10.3390/cimb47110875

**Published:** 2025-10-22

**Authors:** Abass Shaibu Danbaki, Christian Opoku Asamoah, Gideon Okyere Mensah, Bruce Tsri, Tamara D. Busch, Fareed Kow Nanse Arthur, Ishmael Kyei, Lawrence Kobina Blay, Samuel Mensah, Adebowale A. Adeyemo, Azeez Butali, Peter Donkor, Lord Jephthah Joojo Gowans

**Affiliations:** 1Department of Biochemistry and Biotechnology, Kwame Nkrumah University of Science and Technology (KNUST), Kumasi 00233, Ghanacoasamoah6@st.knust.edu.gh (C.O.A.); btsri@st.knust.edu.gh (B.T.);; 2Department of Oral Pathology, Radiology and Medicine, University of Iowa, Iowa, IA 52242, USA; 3University Hospital, Kwame Nkrumah University of Science and Technology (KNUST), Kumasi 00233, Ghana; 4Department of Surgery, Kwame Nkrumah University of Science and Technology (KNUST), Kumasi 00233, Ghana; 5National Human Genomic Research Institute, National Institutes of Health, Bethesda, MD 20892, USA; adeyemoa@mail.nih.gov

**Keywords:** accessory breasts, polymastia, polythelia, multiplex family, whole exome sequencing, *FANCC* gene, secondary findings, comorbidities

## Abstract

Accessory breasts denote the formation of extra breast tissue along the milk line, and are known to be more prevalent among Black and Asian populations, affecting both genders. This first-ever study aimed to determine the genetic aetiology of accessory breasts in a multiplex family, where all female siblings present with bilateral accessory breasts. The study also ascertained secondary findings (SFs) responsible for comorbidities. Clinical data and saliva samples were obtained from all family members. Ultrasound and histopathology confirmed the diagnosis. Whole-exome sequencing was conducted on DNA samples obtained from the saliva, with variant calling conducted utilising the Sentieon workflow. Variant classification was based on American College of Medical Genetics and Genomics guidelines. After segregation analysis, 12 candidate genes emerged. Among these, *PRSS50* and *FANCC* emerged as top candidates, being implicated in breast diseases. However, two variants in *FANCC* (c.360del; p.His120GlnfsTer24 and c.355_358del; p.Ser119IlefsTer24) were selected as the most probable causal variants because of the role of this gene in hereditary breast and ovarian cancer syndromes. The remaining ten genes were reported as potentially accounting for comorbidities segregating with accessory breasts. Reported SFs involve *TTR* and *RYR1*. In conclusion, pathogenic variants in *FANCC* cause familial accessory breasts. These novel observations impact pathophysiology, genetic counselling, and personalised medicine.

## 1. Introduction

Accessory breast tissue is a rare anomaly characterised by extra breast tissue that persists after typical embryonic growth. Occurring in about 2% to 6% of the global human population, it is more prevalent in Black and Asian ancestries, with the Japanese being the most affected. Although it affects both sexes, males experience it the least [1,2]. Anomalous breast tissue refers to both supernumerary breasts and abnormal breast tissue [3,4]. Supernumerary breasts can be found along the milk line, which runs bilaterally and symmetrically from the front axillary lobes to the groin area and the inner portion of the thigh [5]. The armpit is the primary site where extra breast tissue is often detected, although it can also be observed in other body parts, such as the face, back of the neck, and the back and sides of the thigh [6].

Two sets of ectoderm-derived mammary ridges appear during the fourth week of human development [7]. These ridges, often called the “milk line,” extend bilaterally from the front axillary folds to the inguinal folds on the ventral side of the human body. Typically, these ridges disappear, except for the areas where the breasts form [7]. Axillary breast tissue contains most of the components found in normal breast tissue, including the parenchyma, areola, and nipple [8]. Kajava developed a classification system for additional breast tissue in 1915 [2,4]. Class 1, or polymastia, refers to a complete breast with glandular tissue, an areola, and a nipple. A supernumerary breast with only glandular tissue, a nipple, and no areola is classified as Class 2. Class 3 includes an areola with glandular tissue present. Class 4 supernumerary breasts exhibit only glandular tissue. A “pseudomamma,” which comprises only an areola and a nipple, is classified as Class 5. Class 6 is known as polythelia and consists entirely of a nipple, whereas Class 7, termed polythelia areolaris, consists solely of an areola. Class 8 contains only one hair patch, called “polythelial pilosis” [2]. The most common phenotype of accessory breast tissue observed is Class 4 fibroglandular tissue, which often occurs in the axilla.

Most instances of accessory breasts are sporadic, despite the potential for an autosomal dominant inheritance pattern with incomplete penetrance [6]. A person with the condition may not exhibit any symptoms and may be unaware of its development, regardless of where the extra breast tissue is located. Furthermore, accessory breast tissue can be noted during menstruation, pregnancy, or breastfeeding, as it responds to hormonal stimuli in the same manner as normal breast tissue [8].

The identification of supplementary breast tissue has significant treatment implications for each patient. If additional breast tissue is not recognised for what it is, a common variation could be misidentified as another medical defect. Adipose tissue tumours, lymphadenopathy, cutaneous cysts, arterial and venous abnormalities, and cancer are among the most frequently reported probable diagnoses in scientific investigations [9]. According to Goyal et al. (2008) [8], several pathological conditions in normal breast tissue can also arise in accessory breast tissue. These conditions include benign processes such as fibrocystic changes, fibroepithelial lesions, mastitis, atypical ductal or lobular hyperplasia, and true malignancy [8].

The current study, being the first-ever to be conducted among Sub-Saharan Africans, sought to identify the genetic basis of accessory breasts in a multiplex family from Ghana, utilising whole exome sequencing (WES). To the best of our knowledge, this study also pioneers genetic studies on accessory breasts in the Ghanaian population. Importantly, our study is the first to implicate pathogenic *FANCC* gene variants in the aetiology of accessory breasts, suggesting a role of anomalous DNA repair mechanisms in the aetiology of the disease. In addition, we report numerous secondary genomic findings that may underlie comorbidities in the affected family. In essence, the observations from our study are crucial for genetic counselling, the pathophysiology of accessory breast tissues, and targeted therapeutics or personalised medicine.

## 2. Materials and Methods

### 2.1. Study Population

The multiplex family is of Ghanaian descent and was recruited from the Kwame Nkrumah University of Science and Technology (KNUST) Hospital. The affected females presented with bilateral accessory breasts. The Institutional Review Board (IRB) at KNUST granted ethical approval for this cross-sectional study, with approval number CHRPE/AP/120/22. Written informed consent was obtained before subject recruitment, and the family was interviewed using a questionnaire.

The study recruited a family of six individuals: three males and three females. All three female siblings displayed phenotypes of bilateral accessory breasts. The mother was deceased at the time of subject recruitment and could not participate in the study. Thus, only a single multiplex family was recruited. The proband was the first to report to the study site and was clinically evaluated by qualified General Surgeons. Histopathological analysis was conducted on excision biopsies obtained from the bilateral accessory breasts of the proband. The proband then informed the study team of the two other siblings with the same condition, who were also subsequently recruited into the study.

### 2.2. Sample Collection, DNA Processing, and Whole Exome Sequencing

Saliva samples were collected using the Oragene DNA Saliva Collection Kit (DNA Genotek, Stittsville, ON, Canada). DNA processing was conducted at the Human Genetics and Genomics (HuGENE) Laboratory at KNUST. The workflow included DNA extraction and purification, quantification, and quality control checks such as XY genotyping for sex confirmation (Appendix A) [10,11]. DNA extraction and purification from saliva samples followed the Oragene Saliva Protocol [10]. DNA quantification was performed using the Invitrogen QubitTM dsDNA BR Assay kit with the Qubit 4.0 Fluorometer (ThermoFisher Scientific, Waltham, MA, USA). A Nanodrop spectrophotometer was utilised to ascertain the DNA purity by detecting potential contaminants like protein, RNA, or phenol by determining 260/280 and 260/230 absorbance ratios. XY genotyping was carried out as a quality control step to confirm the sexes of study participants [10,11].

Genomic DNA samples were sent to Azenta Life Sciences (USA) for WES at a read depth of 100X (Appendix A). DNA was re-quantified using the Qubit 4 Fluorometer, and the Twist Human Comprehensive Exome library was prepared per manufacturer guidelines. Fragmentation was performed using a Covaris S220, followed by end-repair, adenylation, and adapter ligation. Adapter-ligated fragments underwent PCR amplification, validation via Agilent Tapestation, and hybridization with biotinylated baits. Streptavidin-coated beads captured the hybrid DNA, which was cleaned, amplified, and indexed using Illumina primers.

Sequencing libraries were clustered onto multiple lanes of a flow cell and sequenced using the Illumina HiSeq with a 2 × 150 bp paired-end configuration. The HiSeq Control Software (v2.2.68) (HCS) handled image analysis and base calling, generating binary base call (BCL) files. These were converted to FastQ format and de-multiplexed using Illumina bcf2fastq v2.19, allowing one mismatch in index identification.

### 2.3. Bioinformatic Analysis of Whole Exome Sequencing Dataset

The bioinformatics analysis of the WES dataset included quality control checks with FastQC and Trimmomatic, variant calling with the Sentieon pipeline (Appendix A), and variant prioritisation workflows (Appendix A).

The FastQ files obtained from the Illumina sequencing platform were utilised for bioinformatics analysis (Appendix A). The raw reads underwent quality checks using FastQC 0.11.9 [12] and were trimmed for low-quality bases and adapter sequences using Trimmomatic 0.39 [13]. The reads were aligned to the GRCh38 reference genome using BWA-MEM of Sentieon 202112.01 [14]. After alignment, PCR and optical duplicates were identified, and BAM files were generated. Variant calling, including SNVs and indels, was conducted using Sentieon 202112.01 DNAscope [15,16]. VCF files were normalised through left alignment of indels and splitting multiallelic sites into multiple sites using bcftools 1.13. Transcripts that overlapped were determined for every variant, and the effects of the variants on the transcripts were predicted utilising Ensembl Variant Effect Predictor (VEP) v104. The most severe consequence or impact was selected for every variant for downstream cohort analysis.

Variant prioritisation followed the ACMG guidelines [17]. Variants with an allele frequency of 1% or above were filtered out using databases such as the 1000 Genomes Project [18], Exome Variant Server, dbSNP, and gnomAD [19,20]. Variants below this threshold were assessed for pathogenicity using 11 tools embedded in dbNSFP [21], including SIFT, Polyphen 2, Mutation Taster, Mutation Assessor, MetaRNN, REVEL, MutPred, BaysDel_addAF, ClinPred, CADD, and AlphaMissense (Appendix A). For missense variants, they were classified as pathogenic if at least six of these tools identified them as such. Splice AI evaluated splice variants. CADD was again used to assess loss-of-function variants. The functional impact of variants was further analysed using web-based bioinformatics tools to assess the effects of amino acid changes on protein function, along with expression and phenotypic data from animal models. The Genecard suite, including VarElect and Malacards [22], was used for phenotype- and expression-based variant prioritisation [23]. Segregation analysis determined whether variants co-segregated with the phenotype in affected family members. Finally, pathogenicity was further validated using ClinVar, Online Mendelian Inheritance in Man (OMIM), the Alliance for Genomic Resources, and Human Genome Mutation Database (HGMD) [24,25,26,27].

### 2.4. Sanger Sequencing to Confirm Pathogenic Variants

Sanger Sequencing was carried out to confirm the variants from the WES data obtained from the Next-Generation Sequencing (NGS) platform. The procedure for Sanger sequencing has been published by our group [10]. One primer set was designed to cover the exonic region harbouring the c.360del and c.355_358del variants of the *FANCC* gene. The Primer3 software (v0.4.0) (https://primer3.ut.ee/ (accessed on 5 May 2025)) was used to design the primer set to amplify both variants by adding 500 bp upstream and downstream of the region harbouring the variants. The In Silico PCR platform from the UCSC genome browser was used to determine whether the primer sets annealed to the targeted genomic region. The forward and reverse primers used were 5-TGGCACATTCAGCATTAAACAT-3 and 5-TTGTTTCATAGAGACCACCCC-3, respectively, with an amplicon size of 271 base pairs. A total of 4 ng/μL of DNA in a 10 μL reaction volume was used for polymerase chain reaction (PCR). The PCR conditions have been published [10].

The PCR products were run on a 2% agarose gel, along with size markers at 100 A and 200 V for 20 min to confirm the success of the PCR run. Ultraviolet (UV) light was used to view the gel electrophoretic product. Successfully amplified PCR products were sequenced using an ABI 3730XL DNA sequencer based on Sanger sequencing technology at Functional Biosciences, Madison, Wisconsin (https://functionalbio.com/ (accessed on 10 August 2025)).

Chromatograms from the Sanger sequencing platform were transferred to a Unix workstation, PHRED (http://www.phrap.org/phredphrapconsed.html, (accessed on 10 August 2025) v.0.961028) utilised for base calling, PHRAP (http://www.phrap.org/, (accessed on 10 August 2025) v.0.960731) employed to assemble bases, POLYPHRED (http://droog.gs.washington.edu/polyphred/, (accessed on 10 August 2025) v. 0.970312) subsequently used to scan bases, and CONSED (http://www.phrap.org/consed/consed.html, (accessed on 10 August 2025) v. 4.0) used for viewing reads. Reads were subsequently aligned to the GRCh38 human reference genome using the “Blat” function of UCSC Genome Browser (https://genome.ucsc.edu/ (accessed on 10 August 2025)).

### 2.5. Gene Expression and Interactome Analyses

The expression profiles of the two top candidate genes from bioinformatics analyses of WES datasets, *FANCC* and *PRSS50*, were ascertained using bulk tissues gene expression data in GTEX (https://www.gtexportal.org/home/gene/FANCC, (accessed on 17 July 2025) 2024). The interactions of these two genes with other genes were deciphered employing STRING (https://string-db.org/cgi/network?taskId=bTGzi0bXr9XL&sessionId=bUkeU0Kv11w6, (accessed on 17 July 2025) 2024).

## 3. Results

### 3.1. Clinical Characteristics of the Accessory Breast Phenotype in the Multiplex Family

Bilateral accessory breasts were observed in all three female siblings but not in their male siblings or the father (Figure 1A). However, the mother was deceased at the time of recruitment and was unavailable for sample collection and family history. The oldest sibling (27 years), Parity 1, noticed bilateral accessory breast (2.5 cm in the right and 3 cm in the left) in both axillae since age 13. During pregnancy, the masses increase in size; they decrease in size after lactation.

The second sibling (25 years) with Gravida and Parity 0 became aware of her bilateral accessory breast at age 12. The mass of the left accessory breast measured approximately 2 cm, while the right measured about 2.5 cm. The masses were occasionally painful, usually during her menstrual cycle, but subsided after. She noticed an enlargement in the accessory breasts at the age of 25. The mass measured roughly 4 × 4 cm in the left and right axillae.

The Proband (23 years) also had bilateral accessory breasts in the axilla. The mass was discovered at 14 years, measuring just 2 cm in the right axilla and 3 cm in the left. Upon physical examination at age 23, the mass was non-tender and measured about 4 cm in both axillae. Each mass was not attached to the overlying skin, showed no hyperpigmentation or palpable axillary lymph nodes. Routine laboratory analysis of the proband was normal. Ultrasound of the breast suggested bilateral accessory breast tissue (Figure 1B,C). The proband had an excision biopsy of both accessory breasts under general anaesthesia (Figure 1D–F) for histopathological examination. The two tissue samples were submitted for histopathological analysis, fixed in 10% buffered formalin. The larger nodule weighed 146.0 g and measured 10 × 7.0 × 5.0 cm. The smaller weighed 94.0 g and measured 8.0 × 6.0 × 0.4 cm. The cut surfaces of the tissues showed mostly yellow fat with focal areas of grey fibrous tissue, and the skin surfaces appeared normal. The microscopic anatomy of both tissue samples revealed breast lobules and ductal structures. Observation of fibrous tissue and biopsy showed no atypia. Sections of the nodule exhibited a thin, well-defined mass encased in a delicate fibrous capsule. The lesional area contained both epithelial and stromal components. The stroma comprised bland spindle cells, whereas the epithelial ducts showed no atypia. The overall morphology was characteristic of a fibroadenoma in accessory breast tissue.

### 3.2. Detection of Etiologic Variants for Accessory Breast Phenotype

The paired-end reads generated at a read depth of 100X had an average mean quality score of 35.85, suggesting a base calling accuracy of over 99%. More than 93% of the reads had a mean quality score above 30 (Appendix A). More than 99% of the unique reads aligned to the GRCh38 human reference genome in each sample. Furthermore, over 97% of the exons were sequenced with a coverage read depth of at least 20 reads, indicating uniformity and completeness of the coverage (Appendix A). About 40.4% of the variants were synonymous variants, while missense variants accounted for 35.6%. The remaining variants (frameshift, intron, upstream, downstream, splice region, etc.) accounted for the remaining 24%.

The ACMG guidelines [17] on variant classifications were used to prioritise the probable aetiological variant for the accessory breast phenotype in the family. The annotated rare variants with a MAF ≤ 0.01 were checked for pathogenicity using 11 variant effect predictors embedded in dbNSFP [21]. For missense variants, pathogenicity was established if at least six predictors indicated so. The tools employed included ClinPred, MetaRNN, BaysDel_addAF, REVEL, CADD, AlphaMissense, MutPred2, Polyphen-2, MutationAssessor, MutationTaster, and SIFT. Furthermore, SpliceAI was utilised to determine pathogenicity for splice site variants. CADD and REVEL scores were used to assess loss-of-function variants. These analyses revealed 129 probable pathogenic variants in the proband, comprising 70 missense variants and 59 loss-of-function (LoF) variants, with 9 being novel variants. Segregation analyses were subsequently conducted to ascertain variant(s) that segregated with the accessory breast phenotype in the family.

After analyses using the ACMG guidelines, variants in 12 genes were selected. However, variants in two candidate genes, *PRSS50* and *FANCC* (Table 1, Appendix A), were prioritised as top candidates for the accessory breast phenotype in the family because these two genes have previously been implicated in breast diseases such as breast cancer. In gnomAD (https://gnomad.broadinstitute.org (accessed on 14 October 2025)), the *PRSS50* variant (rs145256818) has a minor allele frequency (MAF) of 0.002026 among Africans. Moreover, the MAF of the two *FANCC* variants, rs766909460 and rs750003253, among Africans in the same database, are 0.0007342 and 0.0007344, respectively. This suggests that these variants are very rare among populations of African descent. *FANCC* is a DNA repair protein that participates in post-replication repair and has a checkpoint function in the cell cycle. This gene has been associated with hereditary breast and ovarian diseases [28], making it the most probable candidate for the accessory breast phenotype in the family. Figure 2 depicts the bulk tissue expression patterns of *FANCC* and *PRSS50* genes and their interactions with other genes. Both genes are expressed in the breasts. However, interactome analysis demonstrated that *FANCC* interacts with essential genes involved in inherited breast cancer syndromes such as *BRCA1*. These evidences support the *FANCC* variants as the most probable causative genetic factor for the familial accessory breast phenotype in the study family. As a quality control step, the *FANCC* variant from the WES analysis was confirmed using Sanger sequencing. The chromatograms from the Sanger sequencing are presented in Figure 3. This sequencing was carried out using both reverse and forward primer sets.

### 3.3. Rare Genetic Variants for Co-Occurring Conditions in Individuals with Polymastia

Aside from *PRSS50* and *FANCC*, numerous other aetiologic variants in 10 genes were found to segregate with the accessory breast phenotype in the family, even though these genes have not been previously linked to breast diseases. These other rare genetic variants could potentially explain co-occurring conditions in individuals with accessory breasts. These genes included *SLC7A7*, *NDE1*, *DIP2B*, *ADGRG6*, *CHDH*, *OR2W1*, *ACKR2*, *C3orf62*, *OR4Q3*, and *MYO1H* (Table 2). The pathogenicity of these variants was determined using the ACMG guidelines for variant classification mentioned earlier.

### 3.4. ACMG Secondary Finding Genes

The WES dataset was evaluated for secondary findings based on the ACMG list for reporting secondary findings in clinical exome and genome sequencing [29]. After variant prioritisation, no variant in the listed ACMG genes segregated perfectly with the accessory breast phenotype in the family. However, a heterozygous missense variant in the *RYR1* gene was identified in the proband and the oldest affected female sibling (Table 3). The *RYR1* variant is classified as likely benign or a variant of uncertain significance (VUS) in ClinVar. The oldest affected female sibling also had a heterozygous missense variant in the *TTR* gene (Table 3). The *TTR* variant observed in the current study is classified as pathogenic in ClinVar. According to the ACMG guidelines on secondary findings, *RYR1* and *TTR* are associated with malignant hyperthermia and hereditary TTR amyloidosis, respectively. Both conditions are inherited in an autosomal dominant manner, as observed in those harbouring the variants [29]. Interestingly, no aetiologic variant in ACMG secondary findings genes was observed in the other affected female sibling with accessory breast, nor any of the male participants in the family.

In summary, we demonstrate for the first time that very rare pathogenic variants in the *FANCC* gene underlie the aetiology of accessory breasts. This implicates aberrant DNA repair mechanisms and dysregulated epithelial proliferation in the aetiology of this rare disease. We also report secondary genomic findings that may underlie comorbidities segregating with accessory breasts. These observations delineate the exploratory nature of WES and its relevance in holistically characterising aetiologic variants in coding regions of the human genome.

## 4. Discussion

### 4.1. Clinical Presentations and Diagnosis of the Accessory Breasts

Accessory breast disease is an uncommon condition that is usually misdiagnosed as a result of its similarity to other medical conditions. Clinical recognition of this condition is necessary to ensure correct diagnosis and operative management. The genetic aetiology of accessory breasts remains largely undiscovered. This study aimed to determine the genetic aetiology of accessory breasts in a multiplex Ghanaian family of six individuals.

Accessory breast tissue responds to hormonal stimuli like normal breast tissue. It is primarily observed during menarche, pregnancy, or lactation. However, supplementary mammary tissue may not be evident in some individuals until pregnancy [7]. The oldest female sibling in the affected family reported that the supplementary accessory breasts were relatively small when they were first discovered. She noted that the masses increased in size during pregnancy and reduced in size after pregnancy. Hormones such as oestrogen, progesterone, and prolactin influence the growth of both supplementary and regular mammary tissue [30]. During pregnancy, increased oestrogen and progesterone levels produce anatomical and physiological changes in the breast tissue. Oestrogen promotes the growth of the ductal system, as well as the fatty and connective tissues surrounding the ducts of the breast. In addition, progesterone stimulates the formation of milk-producing lobules and glands, as well as the enlargement of blood vessels and other breast cells. Thus, these hormonal changes add to adipose tissue proliferation and ductal elongation. Post-pregnancy, oxytocin and prolactin levels increase, while progesterone levels reduce, resulting in a reduction in breast size, as noticed in the eldest affected sibling [31,32].

The second affected sibling became aware of her bilateral axillary breasts at the age of twelve. She reported that the masses were sometimes painful, particularly during her menstrual cycle, but resolved after menstruation. This condition, termed cyclical mastalgia, is also observed in typical breasts and is a common symptom among women. Previous studies have defined it as breast pain at the onset of menstruation [33]. The pain is sporadic and recurrent, corresponding in timing with the menstrual cycle. It may occur unilaterally or bilaterally and is associated with tenderness, heaviness, and swelling. The condition affects up to 70% to 80% of women during their lifetime [34]. A study [35] attributed breast pain during the luteal phase of menstruation to elevated serum oestrogen levels and decreased progesterone levels, leading to a hormonal imbalance, as far as these two hormones are concerned. This phenomenon, termed oestrogen dominance, occasions ductal growth and swelling from the oestrogen, which is not compensated for by progesterone levels, culminating in tenderness and discomfort in the normal or accessory breast tissues. The oestrogen dominance can also occasion the formation of cancerous or fibrocystic lumps, which can also cause pain in normal or accessory breast tissues. Additionally, sex hormones-binding protein (SHBP) has been connected to water retention in breast tissue, adding to breast pain [34]

The youngest affected sibling (proband) exhibited bilateral accessory breasts in the axilla, in line with studies on polymastia [2]. Clinical presentation varies, and mammography or ultrasound is recommended [36]. The current study made an initial ultrasound diagnosis and confirmed it with histopathology. Histopathological examination of excised tissue revealed predominantly yellow fat with focal fibrous tissue in line with normal breast composition [37]. Microscopic analysis recognised breast lobules, ductal structures, and fibrous tissue, with stromal engagement. Kajava (1915) classified accessory breast tissue into eight categories, with axillary fibroglandular tissues falling under class 4 [2]. The histopathological observations in the proband align with those in the literature, revealing glandular epithelium, connective tissue stroma, and subcutaneous fat [38].

### 4.2. Genetic Aetiology of the Accessory Breast Phenotype in the Multiplex Family

Pathogenic variants in the *PRSS50* and *FANCC* genes have been implicated in breast disease. Variants in *PRSS50*, encoding a proteolytic enzyme, have been observed in breast cancer [39]. It holds back the activin signalling, influencing cell regulation and proliferation by decreasing the expression of p27, also called CDKN1B [40]. It also activates NF-kappa B (NF-κB), a transcription factor in immune responses and tumorigenesis [41]. Overexpression of *PRSS50* is associated with breast cancer invasion and malignancy [40], but it is not a top candidate for accessory breast aetiology [42]. Based on these previous reports, the *FANCC* variants were chosen over those of the *PRSS50* to underlie the accessory breast in the multiplex family, considering three observations. Firstly, *PRSS50* has not been previously reported to be associated with hereditary breast diseases, as is the case for *FANCC* [28]. Secondly, *FANCC* interacts with very important genes such as *BRCA1* (Figure 2), which is a well-established gene associated with breast neoplasms. Lastly, loss-of-function (LoF) variants, similar to those observed in *FANCC* in the current study, have been reported in individuals with breast neoplasms [43,44]. *FANCC* is involved in DNA repair, cell cycle checkpoint regulation, and chromosomal stability. It also suppresses cytokine-induced apoptosis via STAT1 activation in response to cytokines and growth factors [45]. *FANCC* is regulated by *TP53*, which binds to its promoter to stimulate transcription [46]. Murine models of the *FANCC* gene display phenotypes such as hypersensitivity to DNA crosslinking agents, spontaneous chromosomal breakage, and loss of germ cells, which results in reduced fertility [47]. *FANCC* is involved in several pathways, including the Fanconi anaemia pathway, cytokine signalling, protein kinase R (PKR)-mediated signalling, and transcription regulation by *TP53* [48]. Variants in *FANCC* are linked to hereditary breast and ovarian cancer syndromes, making it a key candidate gene for accessory breast tissue in the family studied. The current study discovered two pathogenic frameshift deletions: c.360del (p.His120GlnfsTer24, CADD score = 24.6) and c.355_358del (p.Ser119IlefsTer24, CADD score = 27.7), Table 1. The two variants segregated perfectly with the accessory breast phenotype within the family, further supporting the pathogenicity of these variants. These variants are classified as pathogenic in ClinVar and are associated with hereditary neoplastic syndromes, increasing susceptibility to benign and/or malignant neoplasms.

A previous study on actionable mutations in various non-BRCA cancer-associated genes found deletions in similar regions of the *FANCC* gene (c.355_360delTCTCATinsA) in Black/African American women with a familial history of breast cancer [43]. Pathogenic and likely pathogenic variants in *FANCC* were also reported in individuals with breast and ovarian cancer [28]. A study [44] in an Australian population examined the familial predisposition to breast and ovarian cancer in multiple multi-generational breast cancer families without *BRCA1/2* mutations. Two protein-truncating variants in *FANCC* (c.535C>T, [p.Arg179*] and c.553C>T, [p.Arg185*]) were found in 15 *BRCA1/2*-negative families at high risk of breast cancer. This suggests a predisposing role for *FANCC* variants in breast cancer [44]. A similar study conducted in a Chinese cohort with familial breast and/or ovarian cancer that lacked *BRCA1/2* mutations observed a deleterious variant (c.339G>A, W113X) in *FANCC* [49]. However, in a large-scale case–control study in a European population genotyped two truncating variants (p.R185X and p.R548X) of *FANCC* using microarray, as both variants are known to cause disease in the population. Although the variants were detected in approximately 50 individuals, there was no evidence of an association between these variants and the risk of breast cancer compared to *BRCA1* and *BRCA2* mutations [50]. In a nutshell, given the involvement of *FANCC* in DNA repair [45] and epithelial proliferation control [46], we postulate that LoF variants may lead to abnormal mammary tissue development along the embryonic milk line, potentially leading to the formation of accessory breasts.

### 4.3. Other Likely Pathogenic Variants Segregating with the Accessory Breasts

Accessory breasts can be observed as part of pleiotropic syndromes or as individual traits. Studies have reported comorbidities in affected individuals. Based on this, we hypothesised that the variants in the 10 other genes could contribute to conditions that co-occur with accessory breast, but not necessarily the accessory breast phenotype in the family (Table 2). Six missense variants (occurring in *SLC7A7*, *NDE1*, *DIP2B*, *ADGRG6*, *OR2W1*, and *CHDH*), one stop-loss variant (in *MYO1H*), and three frameshift variants (observed in *ACKR2*, *C3orf62*, and *OR4Q3*) segregated perfectly with the accessory breast phenotype in all affected individuals. These genes are involved in crucial biological processes or pathways, including neurodevelopment (*NDE1* and *DIP2B*), multi-system development (*ADGRG5*), immune functions (*ACKR2* and *MYO1H*), fertility (*C3orf62*), olfactory sensory perception (*OR4Q3* and *OR2W1*), and metabolism of certain substrates (*CHDH* and *SLC7A7*). *NDE1* is essential for brain function due to its role in cerebral cortex development [51]. Variants of *NDE1* have been associated with neurodevelopmental disorders, particularly microcephaly related diseases of cortical development [52]. Similarly, *DIP2B* plays a crucial role in neuronal cell growth, with variants in this gene linked to neurodevelopmental diseases [53]. *ADGRG5* is a receptor protein vital for tissue and organ function, including the heart, ear, and sciatic nerve [52]. Variants of this gene have been associated with skeletal defects, intellectual disabilities, and lung disorders [54]. *ACKR2* encodes a chemokine receptor, thus interacting with various inflammatory chemokines in the human body. Variants of this gene have been associated with multiple developmental disorders [55]. The protein encoded by *MYO1H* is pivotal to cellular defence mechanisms such as phagocytosis [56]. Variants of this gene have been linked to central hypoventilation syndromes [57]. *C3orf62* is crucial for spermatogenesis and male fertility, although a recent study has associated a mutation in this gene with myofibroma [58]. *OR4Q3* and *OR2W1* encode olfactory receptors responsible for the sense of smell in humans. Although diseases associated with these genes have been reported less frequently, a recent study linked mutations in the *OR4Q3* gene to glioblastoma [59]. Additionally, renal-urological and cardiovascular malformations can co-occur with accessory breast tissue [2]. The current study identified a variant in *CHDH*, a mitochondrial enzyme involved in choline metabolism [60]. Variants within this gene have been linked to metabolic disorders and, in severe cases, can lead to liver damage, as well as tumour prognosis [61]. Variants in *SLC7A7* are associated with lysinuric protein intolerance. Patients with this condition display various clinical manifestations affecting multiple organs. Severe complications include growth retardation, along with lung and renal malformations [62].

### 4.4. Clinically Actionable Secondary Findings Observed in Family Members

In addition to ascertaining the primary genetic cause of accessory breasts, the current study also aimed to decipher gene variants linked to secondary findings according to ACMG guidelines [30]. A pathogenic missense variant (c.9355C>T, [p.Arg3119Cys]) in *RYR1* was identified in the proband and the eldest affected female sibling. *RYR1* encodes a ryanodine receptor in skeletal muscle and is associated with autosomal dominant malignant hyperthermia [30]. This condition is a musculoskeletal disorder that triggers a potentially fatal hypermetabolic response to stressors or anaesthetic agents. Studies have noted that the malignant hyperthermia phenotype varies significantly depending on the *RYR1* variants. More severe phenotypes are observed in variants occurring at relatively conserved sites in the protein [63]. The current study also observed a pathogenic variant in *TTR* in the oldest female sibling affected by accessory breasts. Variants in *TTR* have been associated with autosomal dominant hereditary *TTR* amyloidosis [30]. A phenotype associated with the heart results from the accumulation of misfolded TTR protein in the organ, thereby impacting its function [64,65]. The *TTR* variant observed in the current study has been classified in ClinVar as pathogenic and is associated with congenital heart malformations.

### 4.5. Limitations of the Study

A limitation of this study is that the observation is limited to a single multiplex family, and more families are required to generalise these observations. The difficulty in distinguishing accessory breasts from other breast malformations may account for the underreporting of the condition, which explains the small sample size. Furthermore, additional functional genomics data on the implicated variants are needed to validate the role of the *FANCC* gene in the aetiology of accessory breasts in the affected multiplex family. However, in the current study, several strategies, including gene expression and interactome studies, were employed to decipher the role of *FANCC* in tissues associated with accessory breasts. Future studies involving larger cohorts and functional assays are needed to confirm the causal role of *FANCC* and to clarify gene-environment interactions underlying accessory breast development.

## 5. Conclusions

Although accessory breast anomaly is a rare anomaly worldwide, this study recruited a multiplex family of six, with all three females exhibiting a bilateral accessory breast phenotype. Among the 12 candidate genes that perfectly segregated with the phenotype in all three affected females, the *FANCC* gene, a DNA repair protein, has been implicated in hereditary breast and/or ovarian disease, making it a top candidate gene underlying the aetiology of the accessory breast phenotype in the family. Pathogenic variants in other genes, such as *SLC7A7* and *CHDH*, participate in key biological processes, including neurodevelopment, metabolism of certain substances, multi-system development and immune function, which may predispose affected individuals to additional comorbidities. Following ACMG guidelines on secondary findings, heterozygous variants in the *RYR1* gene could potentially predispose the two females with the accessory breast phenotype to autosomal dominant malignant hyperthermia. A variant of the *TTR* gene may also predispose the eldest affected sibling to autosomal dominant hereditary *TTR* amyloidosis. These findings suggest a potential role for *FANCC*-mediated DNA repair dysfunction and dysregulation of epithelial proliferation in accessory breast formation and highlight the importance of integrating genomic analysis into the diagnosis of rare developmental anomalies. Functional genomics experiments and larger cohorts are needed to confirm the observations in this study.

## Figures and Tables

**Figure 1 cimb-47-00875-f001:**
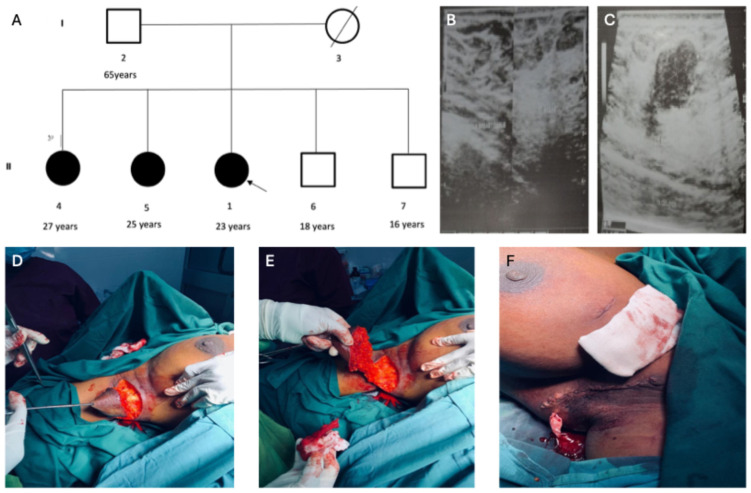
Pedigree of the affected family and clinical presentations of the proband. (**A**): Pedigree of the affected multiplex family. (**B**,**C**): The proband, after an ultrasound examination, was diagnosed with bilateral accessory breasts. (**D**–**F**): Excision biopsy of the accessory breast tissue in the proband for samples for histological examination.

**Figure 2 cimb-47-00875-f002:**
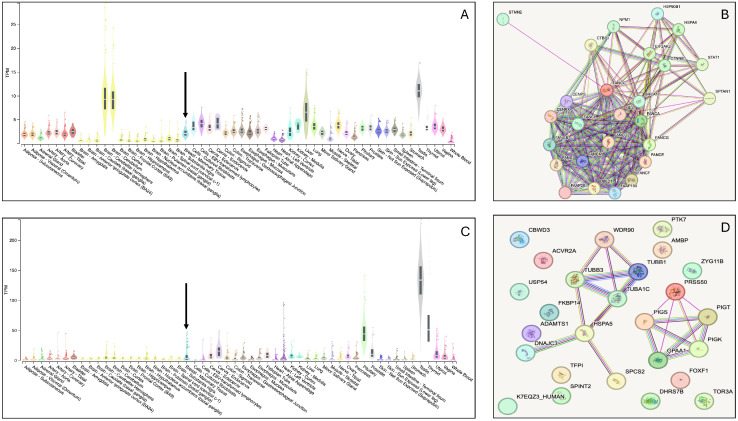
Bulk tissue expression and interactome of the *FANCC* and *PRSS50* genes. (**A**): Bulk tissue expression pattern of *FANCC*. The arrow indicates its expression in the mammary tissue (https://www.gtexportal.org/home/gene/FANCC, (accessed on 17 July 2025) 2024). (**B**): depicts the STRING interaction of the *FANCC* gene, showing its interaction with essential breast cancer-related genes like *BRCA1* (https://string-db.org/cgi/network?taskId=bTGzi0bXr9XL&sessionId=bUkeU0Kv11w6, (accessed on 17 July 2025) 2024). (**C**): Bulk tissue expression pattern of *PRSS50* (https://www.gtexportal.org/home/gene/PRSS50 (accessed on 17 July 2025)), demonstrating expression of the gene in the breast, indicated by an arrow. (**D**): Interactome for *PRSS50*.

**Figure 3 cimb-47-00875-f003:**
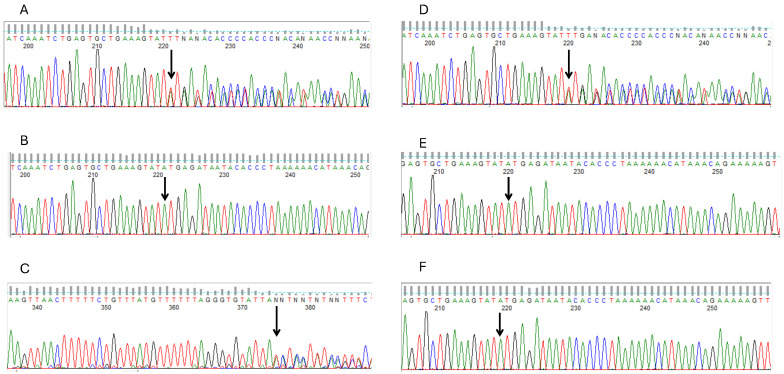
Chromatograms depicting the frameshift variants in the affected individuals. The Sanger sequencing was carried out using both forward and reverse primers. The chromatograms were generated from Sanger sequencing using the reverse primer set, except for (**C**), where that of the forward primer is shown, because the reverse one was not very clear. The arrows indicate the position of the variants. Panels (**A**,**C**,**D**) are the chromatograms from the three affected females, whereas panels (**B**,**E**,**F**) are from the unaffected males in the family. (**A**): 20222001_1, (**B**): 20222001_2, (**C**): 20222001_4, (**D**): 20222001_5, (**E**): 20222001_6, (**F**): 20222001_7.

**Table 1 cimb-47-00875-t001:** Rare genetic variants associated with the accessory breast phenotype in the family.

Genomic Coordinate	ETI	Gene Name	Ref^#^	Alt^#^	Genotype of Affected Individuals	HGVSc	HGVSp	NTP
Chr3:46714301 (rs145256818)	ENST00000460241	*PRSS50*	G	A	G/A	c.671C>T	p.Pro224Leu	7
Chr9:95172132 (rs766909460)	ENST00000289081	*FANCC*	TA	T	TA/T	c.360del	p.His120GlnfsTer24	1
Chr9:95172134 (rs750003253)	ENST00000289081	*FANCC*	TGAGA	T	TGAGA/T	c.355_358del	p.Ser119IlefsTer24	1

All transcripts indicated for the genes are canonical transcripts. All variants were absent in unaffected members. ETI: Ensemble Transcript ID, Ref^#^: Reference allele, Alt^#^: Alternate allele, NTP: Number of tools predicting pathogenicity, HGVSc: Human Genome Variation Society coding sequence, HGVSp: Human Genome Variation Society protein sequence.

**Table 2 cimb-47-00875-t002:** Rare genetic variants that may be responsible for comorbidities.

Genomic Coordinate	ETI	Gene Name	Ref^#^	Alt^#^	Genotype of Affected Individuals	HGVSc	HGVSp	NTP
Chr14:22813019 (rs764284986)	ENST00000397532	*SLC7A7*	A	G	A/G	c.380T>C	p.Ile127Thr	9
Chr16:15687426 (Novel)	ENST00000396355	*NDE1*	C	G	C/G	c.438C>G	p.Ile146Met	9
Chr12:50731373 (rs147225936)	ENST00000301180	*DIP2B*	T	C	T/C	c.3646T>C	p.Tyr1216His	8
Chr6:142415804 (rs1204304460)	ENST00000367609	*ADGRG6*	G	A	G/A	c.2678G>A	p.Arg893Gln	6
Chr3:53823630 (rs765283936)	ENST00000315251	*CHDH*	C	T	C/T	c.379G>A	p.Val127Ile	6
Chr3:42865498 (rs577087357)	ENST00000422265	*ACKR2*	ATGGCACC	A	ATGGCACC/A	c.1010_1016del	p.Pro337LeufsTer21	1
Chr12:109447180 (rs200225794)	ENST00000310903	*MYO1H*	T	C	T/C	c.3115T>C	p.Ter1039ArgextTer1	1
Chr3:49276455 (rs868287641)	ENST00000343010	*C3orf62*	CCT	C	CCT/C	c.416_417del	p.Glu139GlyfsTer20	1
Chr14:19747835 (Novel)	ENST00000642117	*OR4Q3*	CATGAACCCCCAGCT	C	CATGAACCCCCAGCT/C	c.436_449del	p.Asn146ProfsTer32	1
Chr6:29044316 (rs149813138)	ENST00000377175	*OR2W1*	G	A	G/A	c.860C>T	p.Pro287Leu	7

All transcripts indicated for the genes are canonical transcripts. All variants were absent in unaffected members. ETI: Ensemble Transcript ID, Ref^#^: Reference allele, Alt^#^: Alternate allele, NTP: Number of tools predicting pathogenicity. HGVSc: Human Genome Variation Society coding sequence, HGVSp: Human Genome Variation Society protein sequence.

**Table 3 cimb-47-00875-t003:** Variants observed in ACMG secondary findings genes.

Genomic Coordinate	ETI	Gene Name	Ref Allele	Alt Allele	Genotype	HGVSc	HGVSp	NTP
Chr19:38512366 (rs61739911)	ENST00000359596	*RYR1*	C	T	C/T	c.9355C>T	p.Arg3119Cys	7
Chr18:31595139 (rs1555631393)	ENST00000237014	*TTR*	G	A	G/A	c.220G>A	p.Glu74Lys	11

The transcript indicated for the gene is a canonical transcript. ETI: Ensemble Transcript ID, NTP: Number of tools predicting pathogenicity. HGVSc: Human Genome Variation Society coding sequence, HGVSp: Human Genome Variation Society protein sequence.

## Data Availability

The data presented in this study are available on request from the corresponding author due to ethical reasons.

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
