# Peer review of "Pathogenic *FANCC* Variants Are Associated with Accessory Breasts in a Sub-Saharan African Multiplex Family"

_cimb, 2025, doi:10.3390/cimb47110875_

Round 1
Reviewer 1 Report
Comments and Suggestions for Authors
This paper introduces the first genetic investigation of accessory breast tissue in a Sub-Saharan African multiplex family. Using whole-exome sequencing, it identifies pathogenic FANCC variants that segregate with the accessory breast phenotype, implicating DNA repair dysfunction in its aetiology. The study also reports secondary clinically actionable variants, highlighting the broader health implications of genomic findings in rare developmental anomalies.
- The abstract could be more assertive and highlight novelty and implications.
- The final paragraph of the Introduction could emphasize novelty and genetic significance more clearly.
- In Section 3 (Results) , statistical/quantitative clarity could be enhanced (e.g., percentages, allele frequencies).
- Section 3 could end with a short summarizing paragraph tying the major finding (FANCC variants) to the phenotype.
- In Section 4 (Discussions), some sentences are repetitive or overly descriptive (e.g., hormonal explanations restated multiple times). I suggest condensing by combining repetitive hormonal explanations, focusing instead on how hormonal changes highlight the accessory breast’s functional similarity to normal tissue.
- In Section 4, the transition between PRSS50 and FANCC discussions feels abrupt. I suggest adding a linking sentence to clarify why PRSS50 was initially considered but deprioritized.
- In Section 4, the FANCC discussion is strong but could better connect molecular mechanisms to phenotype formation. I suggest adding a short integrative statement, e.g.: “Given FANCC’s involvement in DNA repair and epithelial proliferation control, loss-of-function variants may lead to abnormal mammary tissue development along the embryonic milk line.”
- Section 4 reads as a gene-by-gene list, reducing readability. I suggest grouping genes by functional similarity or organ system (e.g., neurodevelopmental genes, immune/metabolic genes) to emphasize biological pathways over enumeration. Example grouping:
- Neurodevelopmental genes: NDE1, DIP2B, ADGRG5
- Immune/metabolic genes: ACKR2, CHDH, SLC7A7, MYO1H
- Sensory and fertility-related genes: OR4Q3, OR2W1, C3orf62
- Section4: Good acknowledgment of sample size limitation but could include a forward-looking perspective. I suggest ending with a positive note on future directions, e.g.: “Future studies involving larger cohorts and functional assays are needed to confirm the causal role of FANCC and to clarify gene, environment interactions underlying accessory breast development.”
- The Conclusions Section ends abruptly after listing secondary findings. It needs a final sentence that summarizes the broader significance or future implications, e.g.: “These findings suggest a potential role for FANCC-mediated DNA repair dysfunction in accessory breast formation and highlight the importance of integrating genomic analysis into the diagnosis of rare developmental anomalies.”
Comments on the Quality of English Language
Language Comments:
1- In the Abstract: “familial hereditary breast and ovarian cancer syndromes”. I see “familial” and “hereditary” are redundant.
2- In Section 2, some sentences restate details (e.g., “In summary” repeated too often).
3- In Section5, minor stylistic and grammatical refinements: “Accessory breast” should be “accessory breast tissue” or “accessory breast anomaly” for precision. “Predispose affected individuals to co-occurring conditions” should be “predispose to additional comorbidities.”
Author Response
REVIEWER 1
This paper introduces the first genetic investigation of accessory breast tissue in a Sub-Saharan African multiplex family. Using whole-exome sequencing, it identifies pathogenic FANCC variants that segregate with the accessory breast phenotype, implicating DNA repair dysfunction in its aetiology. The study also reports secondary clinically actionable variants, highlighting the broader health implications of genomic findings in rare developmental anomalies.
Comment 1: The abstract could be more assertive and highlight novelty and implications.
Response: We are grateful to the reviewer for this important comment. We have revised the abstract to reflect this suggestion. For example, on the “implications” of the study, we state that “These novel observations impact pathophysiology, genetic counseling and personalized medicine”.
Comment 2: The final paragraph of the Introduction could emphasize novelty and genetic significance more clearly.
Response: We are grateful to the reviewer for this critical comment. We have revised the final paragraph of the introduction to reflect this suggestion. For example, we added the following statements: “Importantly, our study is the first-ever to implicate pathogenic FANCC gene variants in the etiology of accessory breasts, suggesting a role of anomalous DNA repair mechanisms in the etiology of the disease. In addition, we report numerous secondary genomic findings that may underlie comorbidities in the affected family. In essence, the observations from our study are crucial for genetic counselling, the pathophysiology of accessory breast tissues, and targeted therapeutics or personalized medicine”.
Comment 3: In Section 3 (Results) , statistical/quantitative clarity could be enhanced (e.g., percentages, allele frequencies).
Response: Thank you to the reviewer for this observation. However, it is difficult to undertake any quantitative statistical analysis of the dataset because there is no comparative group or control, as the study was purely a family-based one. For allele frequency, we stated in section 2.3 that “Variants with an allele frequency 1%, and above were filtered out using databases such as the 1000 Genomes Project [15], Exome Variant Server, dbSNP and gnomAD”. In addition, we have added the following statements to section 3.2: “In gnomAD (https://gnomad.broadinstitute.org), the PRSS50 variant (rs145256818) has a minor allele frequency (MAF) of 0.002026 among Africans. Moreover, the MAF of the two FANCC variants, rs766909460 and rs750003253, among Africans in the same database, are 0.0007342 and 0.0007344, respectively. This suggests that these variants are very rare among populations of African descent”.
Comment 4: Section 3 could end with a short summarizing paragraph tying the major finding (FANCC variants) to the phenotype.
Response: We are grateful to the reviewer for this comment. We have added the following statements to the end of section 3: “In summary, we demonstrate for the first time that very rare pathogenic variants in the FANCC gene underlie the etiology of accessory breasts. This implicates aberrant DNA repair mechanisms in the etiology of this rare disease. We also report secondary genomic findings that may underlie comorbidities segregating with accessory breasts. These observations delineate the exploratory nature of WES and its relevance in holistically characterizing etiologic variants in coding regions of the human genome”.
Comment 5: In Section 4 (Discussions), some sentences are repetitive or overly descriptive (e.g., hormonal explanations restated multiple times). I suggest condensing by combining repetitive hormonal explanations, focusing instead on how hormonal changes highlight the accessory breast’s functional similarity to normal tissue.
Response: The first two paragraphs of section 4.1 describe two different aspects of clinical presentations of accessory breast vis-à-vis hormonal changes. The first paragraph explains how the size of accessory breasts changes with menarche, pregnancy, or lactation, depicting how various hormones influence the size of accessory breasts at different developmental phases. The second paragraph focuses on pain observed in individuals with accessory breasts, and briefly states hormones that may contribute to this cyclical condition. Whereas some hormones are indicated in both paragraphs, this is important in conveying the role of certain hormones in both clinical presentations. We hope the reviewer will kindly appreciate our approach to conveying this vital information.
Comment 6: In Section 4, the transition between PRSS50 and FANCC discussions feels abrupt. I suggest adding a linking sentence to clarify why PRSS50 was initially considered but deprioritized.
Response: We are grateful to the reviewer for pointing this out. We have included the following information under section 4.2 to address the comment: “Based on these previous reports, the FANCC variants were chosen over those of the PRSS50 to underlie the accessory breast in the multiplex family, considering three observations. Firstly, PRSS50 has not been previously reported to be associated with hereditary breast diseases, as is the case for FANCC[28]. Secondly, FANCC interacts with very important genes such as BRCA1 (Figure 2), which is a well-established gene associated with breast neoplasms. Lastly, loss-of-function (LoF) variants, similar to those observed in FANCC in the current study, have been reported in individuals with breast neoplasms [47,48]”.
Comment 7: In Section 4, the FANCC discussion is strong but could better connect molecular mechanisms to phenotype formation. I suggest adding a short integrative statement, e.g.: “Given FANCC’s involvement in DNA repair and epithelial proliferation control, loss-of-function variants may lead to abnormal mammary tissue development along the embryonic milk line.”
Response: We are grateful to the reviewer for this important comment. We have added the following to the end of section 4.2: “In a nutshell, given the involvement of FANCC in DNA repair[43] and epithelial proliferation control[44], we postulate that LoF variants may lead to abnormal mammary tissue development along the embryonic milk line, potentially leading to the formation of accessory breasts”.
Comment 8: Section 4 reads as a gene-by-gene list, reducing readability. I suggest grouping genes by functional similarity or organ system (e.g., neurodevelopmental genes, immune/metabolic genes) to emphasize biological pathways over enumeration. Example grouping: (1) Neurodevelopmental genes: NDE1, DIP2B, ADGRG5, (2) Immune/metabolic genes: ACKR2, CHDH, SLC7A7, MYO1H, and (3) Sensory and fertility-related genes: OR4Q3, OR2W1, C3orf62.
Response: We thank the reviewer for this comment. Prior to the brief discussion of the functions and conditions associated with the individual genes, the following statement has been added: “These genes are involved in crucial biological processes or pathways, including neuro-development (NDE1 and DIP2B), multi-system development (ADGRG5), immune functions (ACKR2 and MYO1H), fertility (C3orf62), olfactory sensory perception (OR4Q3 and OR2W1) and metabolism of certain substrates (CHDH and SLC7A7)”.
Comment 9: Section4: Good acknowledgment of sample size limitation but could include a forward-looking perspective. I suggest ending with a positive note on future directions, e.g.: “Future studies involving larger cohorts and functional assays are needed to confirm the causal role of FANCC and to clarify gene, environment interactions underlying accessory breast development.
Response: Thank you to the reviewer for this comment. We have added the following statement to section 4: “Future studies involving larger cohorts and functional assays are needed to confirm the causal role of FANCC and to clarify gene-environment interactions underlying accessory breast development”.
Comment 10: The Conclusions Section ends abruptly after listing secondary findings. It needs a final sentence that summarizes the broader significance or future implications, e.g.: “These findings suggest a potential role for FANCC-mediated DNA repair dysfunction in accessory breast formation and highlight the importance of integrating genomic analysis into the diagnosis of rare developmental anomalies.”
Response: We are grateful to the reviewer for this comment. We have added the following statement to the conclusion: “These findings suggest a potential role for FANCC-mediated DNA repair dysfunction and dysregulation of epithelial proliferation in accessory breast formation and highlight the importance of integrating genomic analysis into the diagnosis of rare developmental anomalies”.
Comment 11: In the Abstract: “familial hereditary breast and ovarian cancer syndromes”. I see “familial” and “hereditary” are redundant.
Response: “familial” has been deleted, and “hereditary” has been maintained.
Comment 12: Section 2, some sentences restate details (e.g., “In summary” repeated too often).
Response: Overly repeated phrases have been deleted.
Comment 13: In Section 5, minor stylistic and grammatical refinements: “Accessory breast” should be “accessory breast tissue” or “accessory breast anomaly” for precision. “Predispose affected individuals to co-occurring conditions” should be “predispose to additional comorbidities.”
Response: Thank you to the reviewer. The suggested changes have been effected.
Reviewer 2 Report
Comments and Suggestions for Authors
This manuscript used whole-exome sequencing to study the pathogenic FANCC variants and their relationship with accessory breasts. I have the following concerns:
Major comments:
- The figure 2 resolution is too low and unreadable. Please provide better figures.
- In Figure 3, there are a lot of reads marked with “N” after the indicated arrows in 3A, 3C and 3D. I am wondering how these nucleotides were assigned and analyzed in the study.
Minor comments:
- The full name of abbreviations should be provided, for example, “HSV” in Table 1.
Author Response
REVIEWER 2
This manuscript used whole-exome sequencing to study the pathogenic FANCC variants and their relationship with accessory breasts. I have the following concerns:
Comment 1: The figure 2 resolution is too low and unreadable. Please provide better figures.
Response: We thank the reviewer for pointing this out. We have added a new figure with better resolution. However, the reader must zoom in to see the figure's details, as it appears blurred without zooming.
Comment 2: In Figure 3, there are a lot of reads marked with “N” after the indicated arrows in 3A, 3C and 3D. I am wondering how these nucleotides were assigned and analyzed in the study.
Response: Figures 3A, 3C, and 3D are chromatograms from individuals with the FANCC gene deletions that resulted in frameshift mutations. As is typical of frameshift mutations, nucleotides downstream of the deletion are challenging to detect accurately by the variant calling algorithm due to alterations of the reading frame; hence, the “N” is indicated for the nucleotides downstream of the deletion.
Comment 3: The full name of abbreviations should be provided, for example, “HSV” in Table 1.
Response: Thanks to the reviewer for pointing this out. Full names of abbreviations have been given as footnotes for Tables 1 to 3.
Reviewer 3 Report
Comments and Suggestions for Authors
Dear authors please revise the manuscript as per my suggestions. These changes will improve the the quality of the revised manuscript and would be useful for the readers. My comments have been given below,
- The percent match of 27 % can be reduce to less than 20 %. Kindly pay attentions for this throughout the manuscript.
- Figure 1A, please improve visibility of the text and font size.
- Figure 2. Very poor quality. I could not see the font size clearly. Please improve it.
- The reference format is not uniform. Somewhere it is p, or pp for page number or volumes. Please check the journal guideline.
- The conclusion section does not include the limitations, or challenges of this study. In addition, conclusion is very short. Please improve it.
- The oestrogen and progesterone levels may produce anatomical and physiological changes in the breast tissue. What is the major reasons for this and is there any mechanism which can be explained?
- How breast pain during the luteal phase of menstruation to elevated serum oestrogen levels, lowered progesterone levels and hormonal imbalance?.
- The difficulty in distinguishing accessory breasts from other breast malformations may account for the underreporting of the condition, which explains the small sample size. I think should be revised with clear limitations of this study.
- How ADGRG5 which is a receptor protein plays vital role for tissue and organ function? Is it affecting the heart, ear, and sciatic nerve systems?
- Why OR4Q3 and OR2W1 encode olfactory receptors responsible for the sense of smell in humans?
- The grammar can be improved.
Author Response
REVIEWER 3
Dear authors please revise the manuscript as per my suggestions. These changes will improve the quality of the revised manuscript and would be useful for the readers. My comments have been given below.
Comment 1: The percent match of 27 % can be reduce to less than 20%. Kindly pay attentions for this throughout the manuscript.
Response: Aspects of the manuscripts have been paraphrased to reduce the percentage match.
Comment 2: Figure 1A, please improve visibility of the text and font size.
Response: Thanks to the reviewer for pointing this out. We have added a new and clearer image.
Comment 3: Figure 2. Very poor quality. I could not see the font size clearly. Please improve it.
Response: We thank the reviewer for pointing this out. We have added a new figure with better resolution. However, the reader must zoom in to see the figure's details, as it appears blurred without zooming in.
Comment 4: The reference format is not uniform. Somewhere it is p, or pp for page number or volumes. Please check the journal guideline.
Response: Thank you to the reviewer for this comment. However, what the reviewer is pointing out is actually the correct reference format. “p” is used when the reference is referring to a page, whereas “pp” is used when a range of pages is being referenced. We have also thoroughly revised the references to include missing information. Please note that not all references have issue numbers.
Comment 5: The conclusion section does not include the limitations, or challenges of this study. In addition, the conclusion is very short. Please improve it.
Response: We are grateful to the reviewer for this observation. The conclusion has been expanded, and the limitations have been included at the end. The limitations read: “Functional genomics experiments and larger cohorts are needed to confirm the observations in this study”
Comment 6: The oestrogen and progesterone levels may produce anatomical and physiological changes in the breast tissue. What is the major reasons for this and is there any mechanism which can be explained?
Response: Thank you to the reviewer for pointing out the need for mechanistic insight. “Estrogen promotes the growth of the ductal system, as well as the fatty and connective tissues surrounding the ducts of the breast. In addition, progesterone stimulates the formation of milk-producing lobules and glands, as well as the enlargement of blood vessels and other breast cells”. The quoted statements have been added to section 4.1.
Comment 7: How is breast pain during the luteal phase of menstruation related to elevated serum oestrogen levels, lowered progesterone levels and hormonal imbalance?.
Response: We are grateful to the reviewer for requesting this mechanistic insight. The following information has been added to section 4.1: “elevated serum oestrogen levels and decreased progesterone levels, leading to a hormonal imbalance, as far as these two hormones are concerned. This phenomenon, termed oestrogen dominance, occasions ductal growth and swelling from the oestrogen, which is not compensated for by progesterone levels, culminating in tenderness and discomfort in the normal or accessory breast tissues. The oestrogen dominance can also occasion the formation of cancerous or fibrocystic lumps, which can also cause pain in normal or accessory breast tissues”.
Comment 8: The difficulty in distinguishing accessory breasts from other breast malformations may account for the underreporting of the condition, which explains the small sample size. I think it should be revised with clear limitations of this study.
Response: We thank the reviewer for this comment. However, this limitation has been clearly stated in the manuscript. Our limitation was the small sample size. The statement “The difficulty in distinguishing accessory breasts from other breast malformations may account for the underreporting of the condition, which explains the small sample size”, was added as a possible reason for the low reported prevalence of accessory breasts in the study population. Expanding the quoted statement would have been warranted if the focus of our study was on the diagnosis and prevalence of accessory breasts, as well as the reasons they are underdiagnosed in the study population.
Comment 9: How ADGRG5 which is a receptor protein plays vital role for tissue and organ function? Is it affecting the heart, ear, and sciatic nerve systems?
Response: Thank you to the reviewer for seeking clarification. Many genes, such as ADGRG5, FGFR2 and SHH, are expressed across multiple organs and tissues. Such genes may be crucial for the development and function of these multi-organ systems, and pathogenic variants in these genes may lead to a syndrome with multisystem phenotypes. This is the rationale for the descriptions of ADGRG5 in the manuscript.
Comment 10: Why OR4Q3 and OR2W1 encode olfactory receptors responsible for the sense of smell in humans?
Response: Thank you to the reviewer for seeking clarification. OR4Q3 and OR2W1 encode receptor proteins that are expressed in the nasal cavity. These proteins bind to odour molecules to enhance the perception of smell. These genes are discussed in this context in the manuscript. However, we chose not to belabour the functions of these genes because they are not the focus of the study.
Comment 11: The grammar can be improved.
Response: We have thoroughly reviewed the manuscript and improved the grammar where necessary.
Round 2
Reviewer 1 Report
Comments and Suggestions for Authors
Thanks for addressing may comments in the revised manuscript.
Reviewer 2 Report
Comments and Suggestions for Authors
Thank you for addressing my comments. I have no further concern.
Reviewer 3 Report
Comments and Suggestions for Authors
It is acceptable in present form now.